# ALPHADRIVE: UNLEASHING THE POWER OF VLMS IN AUTONOMOUS DRIVING VIA RL AND REASONING

## ABSTRACT

OpenAI o1 and DeepSeek R1 achieve or even surpass human expert-level performance in complex domains like mathematics and science, with reinforcement learning (RL) and reasoning playing a crucial role. In autonomous driving, recent data-driven end-to-end models have greatly improved planning performance but still struggle with long-tailed problems due to the inherent data imbalance. Some studies integrate vision-language models (VLMs) into autonomous driving, but they typically rely on pre-trained models with simple supervised fine-tuning (SFT) on driving data, without further exploration of training strategies or optimizations specifically tailored for planning. In this paper, we propose AlphaDrive, a RL and reasoning framework for VLMs in autonomous driving. AlphaDrive introduces four planning-oriented RL rewards based on Group Relative Policy Optimization (GRPO) and employs a two-stage planning reasoning training strategy that combines SFT with RL. As a result, AlphaDrive significantly improves both planning performance and training efficiency compared to using only SFT or without reasoning. Moreover, we are also excited to discover that, following RL training, AlphaDrive exhibits some emergent multimodal planning capabilities, which is critical for improving driving safety and efficiency. To the best of our knowledge, AlphaDrive is the first to integrate GRPO-based RL with VLMs in the context of autonomous driving. Code will be released to facilitate future research.

## 1 INTRODUCTION

Autonomous driving has witnessed rapid advances in recent years, with end-to-end autonomous driving emerging as one of the most representative models (Hu et al., 2023; Jiang et al., 2023; Chen et al., 2024a; Prakash et al., 2021; Liao et al., 2024). They take sensor data as input and leverage learnable neural networks to plan the vehicle's future trajectory. Benefiting from large-scale driving demonstrations, end-to-end models continuously improving their planning capabilities by expanding training data and increasing model parameters.

However, due to their black-box nature and lack of common sense, end-to-end models still face significant challenges when handling complex and long-tail driving scenarios. For instance, consider a situation where the vehicle ahead is carrying traffic cones while driving. An end-to-end model may fail to comprehend the relationship between the leading vehicle and the traffic cones, mistakenly assuming that the road ahead is under construction and thus impassable, leading to an incorrect decision to brake. Therefore, relying solely on end-to-end models to achieve high-level autonomous driving remains challenging.

With the success of GPT (Brown et al., 2020), large language models (LLMs) show remarkable comprehension and reasoning abilities (Touvron et al., 2023; Yang et al., 2024). Furthermore, their capabilities have evolved from unimodal text understanding to multimodal vision-language processing. (Liu et al., 2024; Chen et al., 2024c; Bai et al., 2023). The pre-trained knowledge and reasoning abilities of VLMs hold great potential to mitigate the limitations of end-to-end models.

Recently, OpenAI's o1 and o3 (OpenAI, 2024), which incorporate reasoning techniques, achieve performance comparable to or even surpassing that of human experts in fields such as programming. Additionally, DeepSeek R1 (Guo et al., 2025), which leverages reinforcement learning, not only demonstrates "emergent capabilities" and achieves top-tier performance but also requires significantly

lower training costs compared to other models. These advances underscore the immense potential of reasoning techniques and RL in the development of large models.

Existing research on applying VLMs to autonomous driving can be broadly categorized into two directions. The first focuses on leveraging VLMs for the understanding of driving scenes (Sima et al., 2023; Zhou et al., 2024). The second explores the use of VLMs for planning, where some studies treat VLMs as end-to-end systems that process driving images and other inputs to directly predict trajectories (Xu et al., 2023; Chen et al., 2023). However, unlike end-to-end models which are specifically designed for trajectory planning, VLMs operate in a language space and are not inherently suited for precise numerical predictions (Frieder et al., 2024). Consequently, directly employing VLMs for trajectory planning may result in suboptimal performance and even pose safety risks.

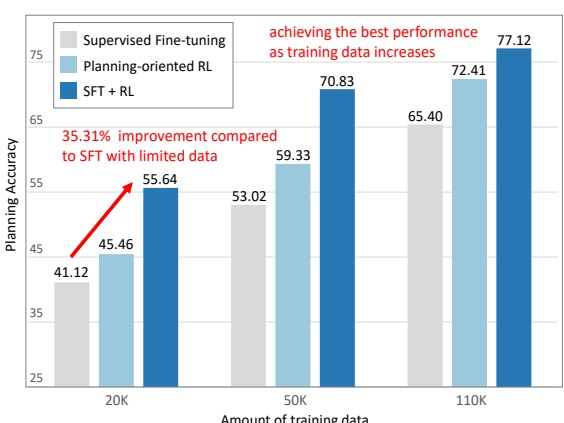

Figure 1: Our planning-oriented RL and two-stage training strategy significantly boost planning accuracy. With limited training data (only 20k samples), it greatly outperforms SFT by 35.31%. As training data increase, AlphaDrive consistently leads in planning performance.

Some studies leverage VLMs for high-level planning by formulating the ego vehicle's future actions in natural language, such as "slow down and turn right" (Jiang et al., 2024). Although this approach circumvents the aforementioned drawbacks, existing works still lack further exploration of training methodologies. Most of them primarily rely on SFT, overlooking the impact of different training strategies on planning performance and the associated training costs.

In this paper, we explore the following question: How can RL and reasoning techniques of VLMs, which have achieved remarkable success in general domains, be applied to autonomous driving to improve planning performance while reducing training costs?

Through preliminary experiments, we find that directly applying existing RL and reasoning techniques to planning results in suboptimal performance. We attribute this to three main factors. First, the reward design in RL for general tasks is not well-suited for planning. For example, in visual object counting, the reward can be simply determined based on whether the model predicts the correct answer. However, in autonomous driving, while high-level planning can be formulated as a multi-class classification problem, the varying significance of different driving behaviors makes it inappropriate to assign equal weights to all actions.

Second, unlike mathematical or counting, the solution of planning are usually not unique. For instance, on an open, straight road, one may choose to maintain a constant speed or accelerate, both of which are valid decisions. Therefore, rigidly assessing whether the model's planning output exactly matches the ground truth in the training data may not be the optimal approach.

Finally, while domains such as mathematics have abundant reasoning data, such as textbooks and solution manuals, autonomous driving lacks readily available datasets that capture the reasoning process. Collecting such data is highly costly and requires extensive manual annotation. As a result, directly applying existing reasoning techniques to planning remains challenging.

To address the aforementioned challenges, this paper introduces AlphaDrive, a VLM-based reinforcement learning and reasoning framework specifically designed for autonomous driving planning. In particular, AlphaDrive employs a RL strategy based on Group Relative Policy Optimization (GRPO) (Shao et al., 2024). Compared to Proximal Policy Optimization (PPO) (Schulman et al., 2017) and Direct Preference Optimization (DPO) (Rafailov et al., 2023), GRPO exhibits better training stability and performance. Furthermore, the group relative optimization strategy is well-suited for planning, as planning often involves multiple valid solutions, making relative optimization across multiple solutions a natural fit. Our experiments show that AlphaDrive exhibits some emergent multimodal planning capabilities, which we think can be attributed to the use of GRPO.

AlphaDrive introduces four GRPO rewards tailored for planning. The first is the planning accuracy reward, which evaluates the consistency between the model's planning actions and the ground truth actions. The second is the action-weighted reward, which assigns different weights to various actions based on their importance to safety. For instance, actions such as braking and steering are critical for safety, so weighting them accordingly helps the model achieve better performance in planning key actions. The third is the planning diversity reward, which encourages the model to generate multiple diverse solutions. This prevents mode collapse and enhances overall planning performance. The last one is the planning format reward, where we define a specific output format and encourage the model to follow it. This ensures more structured outputs and contributes to more stable training.

In addition to RL, we propose a planning reasoning technique which employs a two-stage training strategy based on knowledge distillation, integrating SFT and RL. In the first stage, we leverage a large cloud-based VLM to generate a small yet high-quality dataset, containing planning reasoning processes derived from real driving actions. This dataset is then used to fine-tune our model via SFT, effectively distilling knowledge from the large model. In the second stage, we further refine the model using RL. Introducing SFT as a warm-up stage effectively mitigates hallucinations and instability commonly observed in the early stages of RL, while also enhancing planning performance.

Our contributions are summarized as follows:

- We propose AlphaDrive, a VLM tailored for high-level planning in autonomous driving. To the best of our knowledge, AlphaDrive is the first to integrate GRPO-based RL with VLMs in the context of autonomous driving, significantly boosting both performance and training efficiency.

- AlphaDrive introduces four planning-oriented GRPO rewards: planning accuracy reward, action-weighted reward, planning diversity reward, and planning format reward. These optimized rewards make GRPO more suitable for autonomous driving.

- We propose a two-stage reasoning training strategy based on knowledge distillation, integrating SFT and RL. Our approach achieves better planning performance compared to training with RL alone or without reasoning.

- Extensive experiments and ablations on two datasets validate the superiority of AlphaDrive. Compared to the SFT-trained model, AlphaDrive significantly improves the planning accuracy by 25.52% and, with only 20% of the training data, outperforms the SFT-trained model by 35.31%. We are also excited to discover that, following RL training, AlphaDrive exhibits some emergent multimodal planning capabilities, which is promising for improving driving safety and efficiency.

## 2 RELATED WORK

**Vision Language Models.** The capabilities of large models have greatly expanded from single modality to multi-modalities recently (Brown et al., 2020). Large VLMs (Achiam et al., 2023; Chen et al., 2024b) now demonstrate superior abilities in visual understanding and reasoning. Early works (Alayrac et al., 2022; Li et al., 2022; 2023) attempt to integrate visual models with large language models (LLMs) through attention mechanism and cross-modal contrastive learning. LLaVA (Liu et al., 2024) proposes using vanilla MLP as the connector between. the visual encoder and LLMs, which achieves impressive visual understanding capabilities with relatively limited data. The QwenVL series (Bai et al., 2023; Wang et al., 2024a) continuously improve the visual module, offering better support for high-resolution and dynamic resolution images, while also demonstrate excellent performance in multilingual tasks and spatial perception.

**Reinforcement Learning and Reasoning.** Besides auto-regressive pretraining (Radford et al., 2018), RL and reasoning techniques further enhance the capabilities of large models (Schulman et al., 2017; Wei et al., 2022). For instance, GPT (Achiam et al., 2023) employs RL with Human Feedback (RLHF) (Ouyang et al., 2022), incorporating human preferences into training. By integrating human behavioral preferences, RLHF enables LLMs to generate outputs that align more closely with human preferences. Direct Preference Optimization (DPO) (Rafailov et al., 2023) enhances the model's performance by directly optimizing preference feedback. Building on this, Group Relative Policy Optimization (GRPO) (Shao et al., 2024) introduces group relative optimization, which considers the relative advantages between multiple outputs, further improving training stability and effectiveness. The recent DeepSeek R1 (Guo et al., 2025) experiences an "Aha Moment" during training based on

GRPO, where, without any explicit guidance, the model autonomously allocates more thinking to the problem and re-evaluates its initial approach. This highlights the potential of RL in enabling large models to evolve from mere imitation to emergent intelligence. In our experiments, we are also excited to discover that, after GRPO-based RL training, AlphaDrive demonstrates some emergent multimodal planning capabilities, enabling it to generate multiple reasonable driving plans. We believe it has great potential to improve driving safety and efficiency.

In terms of reasoning, Chain-of-thought (Wei et al., 2022) has demonstrated great performance in solving complex problems by breaking them down and reasoning step by step. OpenAI o1 (OpenAI, 2024), which is based on Chain-of-thought, introduces inference-time scaling. By increasing the computational cost during inference and combining search strategies such as Monte Carlo Tree Search (MCTS) (Świechowski et al., 2023) and Beam Search (Xie et al., 2023), significant improvements have been achieved in areas such as science and programming that require complex reasoning. This also shows that, beyond scaling model parameters and training data, scaling the inference-time computation is also a promising direction for exploration.

**Autonomous Driving Planning.** Planning is the ultimate task of autonomous driving. The earliest planning algorithms are rule-based (Paden et al., 2016; Thrun et al., 2006), which have significant limitations in terms of generalizability and efficiency. Recently, end-to-end models (Hu et al., 2023; Jiang et al., 2023; Chen et al., 2024a; Prakash et al., 2021; Liao et al., 2024; Gao et al., 2025) has gained popularity, where a unified neural network is used to directly output planning trajectories or control signals from sensor data. By leveraging large-scale driving demonstrations, end-to-end models are trained in a data-driven manner, achieving impressive planning performance. However, since end-to-end models are black-box models that lack common-sense and reasoning capabilities, they still struggle to address the long-tailed problems in autonomous driving.

**VLMs and Autonomous Driving.** The common-sense and reasoning capabilities of large models can effectively compensate the limitations of end-to-end models. DriveGPT4 (Xu et al., 2023) employs a VLM that takes front-view videos as input and directly predicts control signals. ELM (Zhou et al., 2024) trains on large-scale, cross-domain video data, showing that diverse data sources can improve VLM performance on driving tasks. OmniDrive (Wang et al., 2024b) introduces sparse 3D tokens to represent driving scenes, which are then input into VLMs for scene understanding and planning.

In addition to the above works that directly apply VLMs for driving, DriveVLM (Tian et al., 2024) combines VLMs with end-to-end models, where VLMs predict low-frequency waypoints and end-to-end models generate high-frequency trajectories. Senna (Jiang et al., 2024) proposes to use VLMs for high-level planning and end-to-end models for low-level trajectory prediction. Several datasets and benchmarks (Sima et al., 2023; Qian et al., 2024) have also been introduced to advance VLM use in autonomous driving. However, most existing work relies on pre-trained models followed by SFT on driving data, lacking exploration of training strategies tailored to planning. Further effort is needed to adapt the impressive RL and reasoning techniques from general tasks to autonomous driving.

## 3 ALPHADRIVE

AlphaDrive is a VLM designed for autonomous driving planning. Unlike previous approaches that rely solely on SFT, we explore the incorporation of RL and reasoning techniques to better align with the unique characteristics of driving planning: (1) the varying importance of different driving behaviors; (2) the existence of multiple feasible solutions; and (3) the scarcity of readily available reasoning data for planning decisions.

We propose four GRPO-based RL rewards tailored for planning, along with a two-stage training strategy that integrates SFT with RL. Our experiments demonstrate that, compared to using SFT alone or training without reasoning, AlphaDrive achieves significant improvements in both planning performance and training efficiency. In the following, we will detail the design of each component.

### 3.1 PLANNING-ORIENTED REINFORCEMENT LEARNING

Current commonly used RL algorithms include PPO (Schulman et al., 2017), DPO (Rafailov et al., 2023), and GRPO (Shao et al., 2024). We ultimately choose GRPO for two key reasons: DeepSeek R1 (Guo et al., 2025) has demonstrated the effectiveness of GRPO in general domains. Compared to

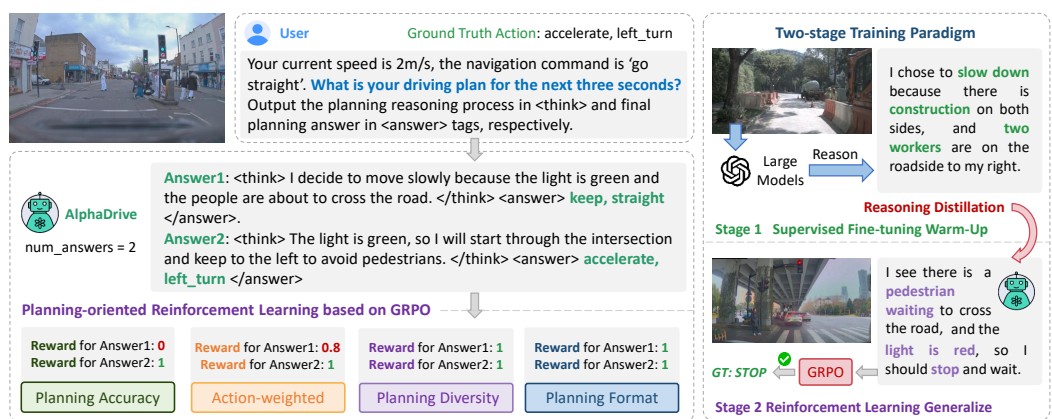

Figure 2: **Overall framework of AlphaDrive.** AlphaDrive is trained using GRPO-based RL, and we design four planning rewards to help the model understand and learn planning. Besides, we propose a two-stage training paradigm, the first stage uses SFT to distill the planning reasoning process from a large model and serves as a warm-up, while the second stage employs RL to explore planning.

other algorithms, GRPO provides higher training stability and efficiency. Moreover, the group relative optimization strategy introduced by GRPO is particularly well-suited for planning, as planning often involves multiple valid solutions, making relative optimization across multiple solutions is a natural fit. Experimental results further confirm that models trained with GRPO exhibit strong planning capabilities.

Given a query $q$, GRPO samples a group of outputs $\{o_1, o_2, \cdots, o_G\}$ from the old policy $\pi_{\theta_{old}}$ and optimizes the new policy $\pi_\theta$ by maximizing:

$$\mathcal{J}_{\text{GRPO}}(\theta) = \mathbb{E}_{q, \{o_i\} \sim \pi_{\theta_{old}}} \left[ \frac{1}{G} \sum_{i=1}^{G} \mathcal{L}_i - \beta \mathbb{D}_{KL}(\pi_\theta || \pi_{ref}) \right], \quad (1)$$

$$\mathcal{L}_i = \min \left( w_i A_i, \text{clip}(w_i, 1 - \epsilon, 1 + \epsilon) A_i \right), \quad (2)$$

where $w_i = \frac{\pi_\theta(o_i|q)}{\pi_{\theta_{old}}(o_i|q)}$, $\epsilon$ and $\beta$ are hyper-parameters, and the advantage $A_i$ is computed using the normalized reward within the group.

### 3.1.1 PLANNING REWARD MODELING

**Planning Accuracy Reward.** In fields such as mathematics or programming, the reward in GRPO can be intuitively determined based on whether the final answer is correct. However, planning is more complex, as it involves both lateral (direction) and longitudinal (speed) components. Furthermore, the set of possible actions is constrained. As a result, we use the F1-Score to evaluate the accuracy of both lateral and longitudinal decisions separately, and assign rewards accordingly.

Initially, we evaluate accuracy by checking whether the model's prediction exactly matches the ground truth. However, due to imperfect format in the model's early training phase, such as discrepancies in case sensitivity or the presence of extraneous outputs, this approach results in poor stability during the early stages of training. We then attempt to extract all the words from the prediction and check whether the ground truth is included among the words. This introduces a new issue where the model sometimes learns shortcut solutions, such as outputting all possible actions, which causes mode collapse. Ultimately, we adopt the F1-score for evaluation, as it not only prevents the model from learning shortcut solutions (where outputting all decisions could result in high recall but low accuracy) but also improves the stability during the early training phase.

**Action-Weighted Reward.** As mentioned above, the importance of different behaviors in planning varies. For instance, decelerating and stopping are more critical for safety than maintaining speed. Therefore, we assign different importance weights to various actions, incorporating them as weighted components in the final reward.

---

**Algorithm 1:** Planning Reward Modeling.

---

**Input:** Planning answers $\mathcal{A}$, Ground Truth action $e$, Action Weights $\mathcal{W}$, Planning Diversity Weight $\theta$
**Output:** Planning Reward $\mathcal{R}$

1 Initialization: Planning Reward $\mathcal{R} \leftarrow \emptyset$, Speed Action Set $\mathcal{S}$, Path Action Set $\mathcal{P}$, Answer Format $\mathcal{F}$
2 # Pytorch-like Code
3 speed_cnt, path_cnt = Counter(), Counter()
4 **for** *ans in $\mathcal{A}$* **do**
5     action_ans = re.search(r"$\mathcal{F}$", ans).group(1).strip()
6     speed_ans, path_ans = **extract_ans**(action_ans, $\mathcal{S}$), **extract_ans**(action_ans, $\mathcal{P}$)
7     speed_cnt.update(speed_ans), path_cnt.update(path_cnt)
8     # Calculate Planning Accuracy Reward
9     speed_acc_R, path_acc_R = **cal_f1_score**(speed_ans, $e$), **cal_f1_score**(path_ans, $e$)
10     # Calculate Action-Weighted Reward
11     speed_weighted_R, path_weighted_R = $\mathcal{W}$[speed_ans], $\mathcal{W}$[path_ans]
12     # Calculate Planning Diversity Reward
13     speed_div_R, path_div_R = $\theta$ **if** speed_cnt[speed_ans] == 1 **else** -$\theta$, $\theta$ **if** path_cnt[path_ans] == 1 **else** -$\theta$
14     # Calculate Planning Format Reward
15     format_R = **check_format**(*ans*, $\mathcal{F}$)
16     # Final Planning Quality Reward
17     speed_R = speed_acc_R * speed_weighted_R + speed_div_R
18     path_R = path_acc_R * path_weighted_R + path_div_R
19     $\mathcal{R}$.append([speed_R, path_R, format_R])
20 **end**
21 Return: $\mathcal{R}$

---

**extrat_ans** will extract substrings that match the specified pattern from the given string. **cal_f1_score** will calculate F1 score given the predictions and ground truth. **check_format** will check whether the given string matches the provided pattern based on regular expression matching.

**Planning Diversity Reward.** Since planning is inherently multimodal, during GRPO-based RL training, the model generates multiple solutions for group relative optimization. In the later stages of training, we observe that the model's outputs tend to converge to the same solution. Our goal is to encourage the model to generate a variety of feasible solutions, rather than merely aligning with the ground truth actions in the training data. To achieve this, we propose the Planning Diversity Reward. When the model's outputs differ, we assign a higher reward; otherwise, we reduce the reward.

**Planning Format Reward.** The last reward is used to regularize the output, making it easier to extract both the reasoning process and the final answer. This approach is inspired by R1. The reasoning process is encapsulated within the `<think></think>` tags, while the planning result is enclosed within the `<answer></answer>` tags. If the final output does not conform to this format, the format reward will be set to 0.

The Planning Accuracy Reward, the Action-Weighted Reward, and the Planning Diversity Reward are combined to compute the Planning Quality Reward. We calculate the Planning Quality Reward separately for speed planning and direction planning. Finally, the Planning Quality Reward and the Planning Format Reward are used to calculate the GRPO loss and update the model parameters. For details about Planning Reward Modeling, please refer to Alg. 1.

### 3.2 REASONING: DISTILLATION FROM LARGE MODELS

Unlike fields such as mathematics or science, which have abundant high-quality reasoning data available for training, the planning process in autonomous driving is difficult to record, and the cost of manual annotation is high. As a result, there is currently no large-scale, readily available planning reasoning dataset. We initially attempt to incorporate reasoning steps directly into the RL training process, but the final results are suboptimal, mainly due to the following shortcomings: (1) insufficient perception of key elements, such as traffic lights; (2) disorganized reasoning process with weak causal relationships; (3) reasoning outputs that are overly lengthy and ineffective.

Therefore, we adopt a more capable cloud-based large model Qwen2VL-72B, to generate high-quality planning reasoning data from a small set of driving clips. Specifically, we provide the model with prompts that include the real driving actions in a given scenario, along with the vehicle's current state and navigation information, prompting the model to generate a concise decision-making process. We

Table 1: High-level planning and reasoning results on the MetaAD validation set. Except for AlphaDrive, which utilizes our proposed training strategy, all other models (Chen et al., 2024b; Wang et al., 2024a; Dubey et al., 2024) are fine-tuned with SFT on the MetaAD training set.

| Method | Train. Strategy | Acc. (%) | Path (F1) ↑ | | | | Speed (F1) ↑ | | | BLEU-4 | CIDEr | METEOR |
|--------|-----------------|----------|----------|------|-------|------|------|------|------|--------|-------|--------|
| | | | straight | left | right | keep | acc. | dec. | stop | | | |
| InternVL2-2B | SFT | 51.07 | 76.13 | 85.16 | 64.60 | 74.77 | 21.88 | 47.66 | 75.81 | 27.89 | 19.73 | 28.26 |
| Qwen2VL-2B | SFT | 55.84 | 82.68 | 80.31 | 70.04 | 75.97 | 34.92 | 55.55 | 72.64 | 24.46 | 23.14 | 34.26 |
| Llama3.2-V-11B | SFT | 58.21 | 85.58 | 84.64 | 79.12 | 74.79 | 35.56 | 58.99 | 76.20 | 32.05 | 21.25 | 37.70 |
| Qwen2VL-7B | SFT | 61.44 | 86.45 | 85.84 | 87.75 | 84.53 | 43.81 | 56.30 | 73.80 | 41.09 | 30.65 | 47.47 |
| AlphaDrive-2B | Ours | **77.12** | **96.62** | **89.83** | **93.25** | **86.80** | **56.33** | **71.40** | **86.63** | **43.54** | **38.97** | **55.23** |

Table 2: End-to-end planning results on the NAVSIM `navtest` split with closed-loop metrics. * denotes incorporating (Liao et al., 2024) as the end-to-end trajectory planning module.

| Method | Reference | NC ↑ | DAC ↑ | TTC ↑ | Comf ↑ | EP ↑ | PDMS ↑ |
|--------|-----------|------|-------|-------|--------|------|--------|
| UniAD (Hu et al., 2023) | CVPR 23 | 97.8 | 91.9 | 92.9 | 100 | 78.8 | 83.4 |
| PARA-Drive (Weng et al., 2024) | CVPR 24 | 97.9 | 92.4 | 93.0 | 99.8 | 79.3 | 84.0 |
| Transfuser (Prakash et al., 2021) | PAMI 23 | 97.7 | 92.8 | 92.8 | 100 | 79.2 | 84.0 |
| DRAMA (Yuan et al., 2024) | arXiv 23 | 98.0 | 93.1 | 94.8 | 100 | 80.1 | 85.5 |
| VADv2 (Chen et al., 2024a) | arXiv 24 | 97.2 | 89.1 | 91.6 | 100 | 76.0 | 80.9 |
| Hydra-MDP (Li et al., 2024) | arXiv 24 | **98.3** | 96.0 | 94.6 | 100 | 78.7 | 86.5 |
| DiffusionDrive (Liao et al., 2024) | CVPR 25 | 98.2 | 96.2 | 94.7 | **100** | 82.2 | 88.1 |
| AlphaDrive-SFT* | Baseline | 98.1 | 96.4 | 95.0 | **100** | 82.4 | 88.3 |
| AlphaDrive* | Ours | **98.3** | **97.6** | **95.4** | **100** | **83.1** | **89.5** |

find that the quality of the generated reasoning process is overall satisfactory. After conducting a manual quality check and filtering out samples with obvious errors, we obtain a batch of high-quality planning reasoning data. Subsequently, our model can improve its planning reasoning ability through knowledge distillation based on this data.

### 3.3 TRAINING: SFT WARM-UP, RL EXPLORATION

RL relies on sparse reward signals, whereas SFT is based on dense supervision, making it more suitable for knowledge distillation. Additionally, we find that relying solely on RL can lead to instability in the early stages of training. Therefore, we use a small amount of data for a warm-up phase based on SFT, followed by RL training with the full dataset. We discover that this approach improves stability in the early stages of training and enhances the model's planning reasoning performance, ultimately leading to better overall planning capabilities.

## 4 EXPERIMENTS

**Dataset.** We adopt two datasets, MetaAD and NAVSIM (Dauner et al., 2024). MetaAD is a large-scale real-world driving dataset which consists of 120k driving clips, each lasting three seconds. It supports multi-sensor data and perception annotations, as well as maintaining a well-balanced distribution across various driving scenarios and planning actions. The dataset is divided into 110k clips for training and 10k clips for validation. As for reasoning, we sample 30k data from the training dataset to generate the planning reasoning data. We conduct ablations on the MetaAD dataset by default.

NAVSIM (Dauner et al., 2024) is a widely used planning benchmark that includes surround-view images from eight cameras, LiDAR point clouds, high-definition maps, and object detection annotations. The dataset supports non-reactive simulations and provides closed-loop evaluation metrics, allowing for comprehensive evaluation of autonomous planning methods. More information about the dataset and ablations can be found in Appendix A due to page limit.

**Training Details.** We use Qwen2VL-2B (Wang et al., 2024a) as the base model. Qwen2VL is currently one of the best-performing open-source models, and it offers a smaller 2B version that better meets the latency requirements for autonomous driving. The model's inputs include a front-view image and a planning prompt, which contains the vehicle's current speed and navigation information.

Table 3: Ablations on the effectiveness of our proposed planning GRPO rewards.

| ID | Base Acc. | Plan. Acc. | Action Weighted | Plan. Diversity | Plan. Format | Acc. (%) | Path (F1) ↑ | | | | Speed (F1) ↑ | | |
|---|---|---|---|---|---|---|---|---|---|---|---|---|---|
| | | | | | | | straight | left | right | keep | acc. | dec. | stop |
| 1 | ✓ | | | | | 42.36 | 69.40 | 64.42 | 59.02 | 62.18 | 23.72 | 47.48 | 62.70 |
| 2 | ✓ | | | | ✓ | 55.71 | 83.19 | 77.34 | 71.65 | 67.37 | 34.07 | 59.87 | 76.56 |
| 3 | | ✓ | | | ✓ | 67.91 | 91.95 | 82.65 | 88.01 | 77.74 | 49.79 | 61.38 | 85.75 |
| 4 | | ✓ | ✓ | | ✓ | 72.20 | 95.93 | 85.39 | 88.80 | 82.54 | 52.64 | 67.60 | **86.76** |
| 5 | | ✓ | | ✓ | ✓ | 69.38 | 92.10 | 80.48 | 85.59 | 84.53 | 49.40 | 64.07 | 83.34 |
| 6 | | ✓ | ✓ | ✓ | ✓ | **77.12** | **96.62** | **89.83** | **93.25** | 86.80 | **56.33** | **71.40** | 86.63 |

Table 4: Ablations on different reasoning training strategies.

| With Reason. | Train. Strategy | Acc. (%) | Path (F1) ↑ | | | | Speed (F1) ↑ | | | BLEU-4 | CIDEr | METEOR |
|---|---|---|---|---|---|---|---|---|---|---|---|---|
| | | | straight | left | right | keep | acc. | dec. | stop | | | |
| ✗ | SFT | 56.97 | 77.76 | 63.69 | 65.07 | 76.22 | 37.11 | 51.99 | 75.72 | - | - | - |
| ✗ | RL | 62.16 | 82.32 | 72.39 | 71.24 | 75.03 | 41.13 | 61.08 | 79.15 | - | - | - |
| ✗ | SFT+RL | 70.73 | 88.04 | 75.75 | 78.79 | 78.60 | 45.00 | 65.92 | 83.52 | - | - | - |
| ✓ | SFT | 65.40 | 92.52 | 71.28 | 68.65 | 81.91 | 36.48 | 59.31 | 71.55 | 37.21 | 34.30 | 47.54 |
| ✓ | RL | 72.41 | 93.16 | 84.24 | 89.32 | **87.58** | 51.19 | 64.70 | 84.07 | 25.14 | 24.58 | 38.10 |
| ✓ | SFT+RL | **77.12** | **96.62** | **89.83** | **93.25** | 86.80 | **56.33** | **71.40** | **86.63** | **43.54** | **38.97** | **55.23** |

The navigation data, consistent with real-world driving, is obtained from sparse navigation points via AMap (similar to Google Maps) and is converted into text form for inclusion in the prompt, such as "Go straight for 100m, then turn right". The training settings follows Qwen2VL, and all experiments are conducted using 16 NVIDIA A800 GPUs.

**Evaluation.** For high-level planning, the evaluation metrics consist of two aspects. First, the accuracy of meta-action planning is measured by calculating the F1-Score for all categories of lateral and longitudinal meta-actions, followed by the overall planning accuracy. Additionally, for planning reasoning, we compute the similarity between the generated planning reasoning process and the annotated reasoning process in the dataset using BLEU-4 (Papineni et al., 2002), CIDEr (Vedantam et al., 2015), and METEOR (Banerjee & Lavie, 2005) scores. In terms of end-to-end planning, we adopt the closed-loop metrics such as PDMS proposed in NAVSIM for evaluation.

**LLM Usage.** We used existing LLMs/VLMs in two ways. As described in Sec. 3.2, Qwen2VL-72B was employed to generate planning-reasoning text for the SFT training data. Additionally, GPT-5 was used for language polishing. The authors take full responsibility for all generated content.

## 4.1 MAIN RESULTS

**High-level Planning.** Tab. 1 presents the performance of AlphaDrive in high-level planning. As shown, AlphaDrive significantly outperforms the other models. Compared to Qwen2VL-7B, the second-best performing model after AlphaDrive, the planning accuracy significantly improves by 25.5%. There is a noticeable enhancement in key decisions such as steering and acceleration/deceleration. Additionally, the quality of planning reasoning is the best among all models, demonstrating the effectiveness of our proposed two-stage RL training and reasoning strategies.

**End-to-end Trajectory Planning.** Besides high-level planning, we further integrate AlphaDrive with an existing end-to-end model to evaluate its contribution to trajectory planning. Specifically, AlphaDrive is first trained on NAVSIM using the same pipeline as the MetaAD dataset. Its high-level decisions are then mapped to high-dimensional features via learnable embeddings and fed as conditional inputs to the DiffusionDrive decoder to generate the final trajectory. As shown in Table 2, the SFT-trained baseline achieves only a marginal improvement, while AlphaDrive, trained with our proposed strategy, achieves the best planning performance, with a PDMS score of 89.5.

## 4.2 ABLATION STUDY

**Planning Rewards.** In Tab. 3, we validate the effectiveness of the proposed planning GRPO rewards. Base Accuracy Reward directly determines the reward based on whether the response exactly matches

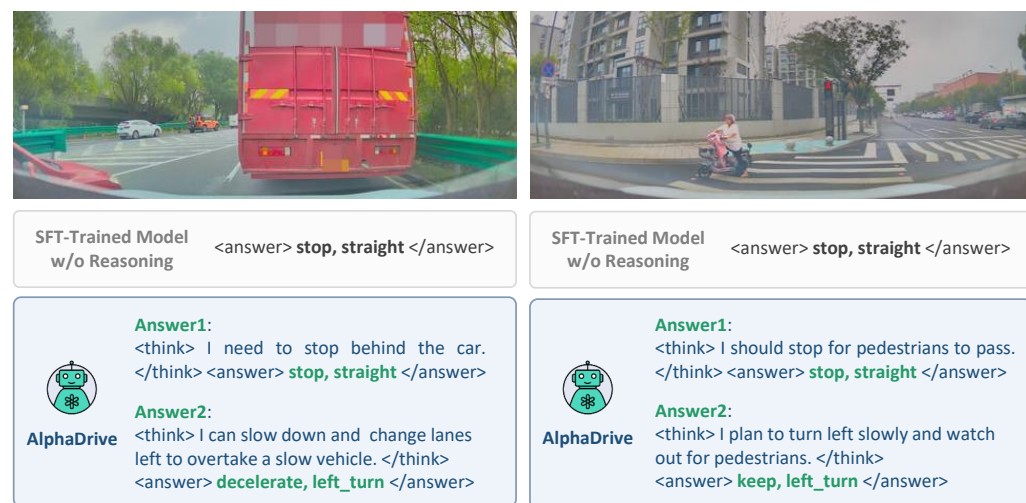

Figure 3: Qualitative results of AlphaDirve. After RL training, AlphaDrive exhibits some emergent multimodal planning capabilities, which has great potential for improving planning performance.

the ground truth, a common approach in general domains. As shown, the model using the Base Accuracy reward lags significantly behind across all metrics (ID 1). The combination with the Planning Format Reward yields a slight improvement. (ID 2). A significant improvement is seen with the adoption of our proposed Planning Accuracy Reward (ID 3). Further enhancement in acceleration/deceleration is achieved by incorporating the Action-Weighted Reward (ID 4). Finally, by combining the Planning Diversity Reward, the best planning performance is achieved (ID 5-6).

**Reasoning Training Strategies.** The ablations of the reasoning training strategies is shown in Tab. 4. Introducing planning reasoning under various training strategies effectively enhances model performance, particularly for complex actions like acceleration and deceleration. This highlights reasoning's impact on decision-making in challenging scenarios. Notably, the RL-trained-only model underperforms in reasoning compared to the SFT-trained one, which we attribute to the limited model size that constrains its perception and reasoning capabilities. Incorporating SFT as a warm-up and using knowledge distillation to learn the reasoning process from a larger model helps mitigate this. Combining SFT and RL yields the best planning reasoning capabilities.

### 4.3 Emergence of Multimodal Planning Capability

Fig. 3 illustrates the multimodal planning capability of AlphaDrive after RL training. In complex scenarios, it can effectively generate multiple feasible solutions. Although SFT-trained model can also generate multiple answers through sampling, its multimodal planning capability remains limited, as shown in our ablation study (Tab. 6). AlphaDrive can be integrated with a downstream action model to dynamically select the optimal solution from multiple options.

## 5 Conclusions and Limitations

In this work, we propose AlphaDrive, a VLM for high-level planning in autonomous driving. Compared to previous models that solely employed the SFT, we explore the integration of advanced RL and reasoning in planning. Specifically, AlphaDrive introduces a planning-oriented RL strategy based on GRPO and further designs a two-stage planning reasoning training paradigm. To the best of our knowledge, AlphaDrive is the first to integrate GRPO-based RL with VLMs in the context of autonomous driving, significantly boosting both performance and training efficiency.

Currently, due to a lack of rich data annotation, AlphaDrive is still unable to output more complex driving behaviors such as lane changes or nudges. Additionally, the current planning reasoning data come from pseudo-labels generated by large models based on ground-truth driving actions, which still suffer from inaccurate perception and a failure to capture key factors. Therefore, further systematic validation is required to improve data quality and verify the performance upper bound.

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

Table 5: High-level planning results on the NAVSIM dataset. ‡ denotes that the model is initialized with weights pre-trained on the MetaAD dataset.

| Method | Acc. (%) | Path (F1) ↑ | | | | Speed (F1) ↑ | | |
| --- | --- | --- | --- | --- | --- | --- | --- | --- |
| | | straight | left | right | keep | acc. | dec. | stop |
| Qwen2VL (Wang et al., 2024a) | 84.68 | 97.23 | 70.02 | 74.14 | 92.83 | 22.78 | 20.99 | 91.12 |
| AlphaDrive | 87.54 | 97.76 | 76.51 | 79.52 | 94.76 | 28.46 | 25.39 | 92.05 |
| AlphaDrive‡ | **92.44** | **98.67** | **83.53** | **82.21** | **95.12** | **45.27** | **53.94** | **95.31** |

Table 6: Abalation on different planning models and multimodal planning results on MetaAD dataset.

| Method | # Samples | Acc. (%) | Path (F1) ↑ | | | | Speed (F1) ↑ | | |
| --- | --- | --- | --- | --- | --- | --- | --- | --- | --- |
| | | | straight | left | right | keep | acc. | dec. | stop |
| ViT-L/14[†] (Dosovitskiy et al., 2020) | 1 | 36.77 | 78.19 | 52.48 | 57.27 | 72.26 | 25.41 | 43.50 | 74.15 |
| ViT-L/14[†] (Dosovitskiy et al., 2020) | 2 | 43.12 | 82.83 | 56.92 | 60.01 | 76.96 | 29.56 | 48.35 | 77.10 |
| Qwen2VL (Wang et al., 2024a) | 1 | 55.84 | 82.68 | 80.31 | 70.04 | 75.97 | 34.92 | 55.55 | 72.64 |
| Qwen2VL (Wang et al., 2024a) | 2 | 58.80 | 87.00 | 84.33 | 73.20 | 77.61 | 38.95 | 59.82 | 76.63 |
| AlphaDrive | 1 | 77.12 | 96.62 | 89.83 | 93.25 | 86.80 | 56.33 | 71.40 | 86.63 |
| AlphaDrive | 2 | **85.39** | **97.44** | **92.19** | **95.94** | **92.74** | **71.58** | **83.75** | **90.72** |

# A APPENDIX

## A.1 HIGH-LEVEL PLANNING RESULTS ON THE NAVSIM DATASET

We further evaluated the planning performance of AlphaDrive on the publicly available NAVSIM dataset (Dauner et al., 2024), as summarized in Tab. 5. The NAVSIM dataset was reformatted into visual question-answering data compatible with AlphaDrive. We compared the performance of three variants: (1) the original Qwen2VL-2B model trained via supervised fine-tuning (SFT) (first row), (2) the model trained using our proposed strategy (second row), and (3) the model first pretrained on the MetaAD dataset and subsequently fine-tuned on NAVSIM using our RL approach (third row). As shown, our training strategy leads to a substantial improvement in planning performance. Furthermore, pretraining on a larger-scale dataset enables the model to generalize effectively across diverse scenarios.

## A.2 MULTIMODAL PLANNING CAPABILITY

We conducted an ablation study with multiple planning generation samples to assess the advantage of AlphaDrive over the SFT-trained model in terms of multimodal planning capabilities. A prediction is deemed correct if at least one of the generated actions matches the ground truth. As illustrated in Tab. 6, when sampling two candidates, AlphaDrive significantly outperforms Qwen2VL-2B trained with SFT, highlighting the effectiveness of our reinforcement learning strategy in improving the model's ability to produce diverse and accurate planning actions.

## A.3 COMPARISON WITH A SUPERVISED PLANNING CLASSIFIER

Since high-level planning can be formulated as a classification task over a finite set of actions, we further evaluated the performance of a simple vision classifier to emphasize the rationale and advantages of employing VLMs for planning. Specifically, we employ ViT-L/14 from CLIP (Radford et al., 2021) as the visual encoder, followed by two MLP classification heads to predict path and speed actions. To ensure consistency with AlphaDrive, we also provide state information such as navigation commands and ego speed to the classification heads. The classifier is trained using a weighted cross-entropy loss, which integrates action-specific weighting into the supervision signal similar to AlphaDrive's GRPO reward design.

As shown in the first two rows of Tab. 6, this vision classifier exhibits poor planning performance, even underperforming the baseline Qwen2VL. These results highlight that, compared to pure vision models, VLMs equipped with commonsense knowledge and reasoning capabilities can more effectively improve high-level planning performance.

Table 7: End-to-end trajectory planning results on the MetaAD dataset.

| Method | Type | Planning L2 (m) ↓ | | | | | |
| --- | --- | --- | --- | --- | --- | --- | --- |
| | | 0.5s | 1s | 1.5s | 2s | 2.5s | 3s |
| VADv2 (Chen et al., 2024a) | E2E | 0.35 | 0.56 | 0.73 | 1.69 | 2.40 | 3.15 |
| Qwen2VL (Wang et al., 2024a) | VLM | 0.73 | 1.14 | 1.70 | 2.85 | 3.67 | 4.54 |
| AlphaDrive-SFT | VLM+E2E | 0.32 | 0.50 | 0.63 | 1.43 | 2.15 | 2.68 |
| AlphaDrive | VLM+E2E | **0.28** | **0.44** | **0.53** | **1.27** | **1.86** | **2.39** |

Table 8: Ablations on the training dataset size.

| Train. Data | Train. Strategy | Acc. (%) | Path (F1) ↑ | | | | Speed (F1) ↑ | | | BLEU-4 | CIDEr | METEOR |
| --- | --- | --- | --- | --- | --- | --- | --- | --- | --- | --- | --- | --- |
| | | | straight | left | right | keep | acc. | dec. | stop | | | |
| 20k | SFT | 41.12 | 56.15 | 36.72 | 35.59 | 40.63 | 17.14 | 16.74 | 19.19 | 27.18 | 15.42 | 31.17 |
| 20k | RL | 45.46 | 69.28 | 59.42 | 51.91 | 56.93 | 30.82 | 37.71 | 30.94 | 20.33 | 11.01 | 23.09 |
| 20k | SFT+RL | 55.64 | 68.25 | 64.06 | 56.87 | 58.61 | 45.19 | 53.68 | 44.09 | 32.84 | 17.02 | 35.93 |
| 50k | SFT | 53.02 | 73.74 | 62.45 | 65.43 | 70.07 | 33.83 | 38.94 | 53.96 | 34.48 | 26.83 | 42.85 |
| 50k | RL | 59.33 | 77.69 | 68.55 | 73.82 | 77.05 | 40.72 | 45.20 | 57.06 | 22.37 | 16.81 | 25.81 |
| 50k | SFT+RL | 70.83 | 82.30 | 78.05 | 82.17 | 84.80 | 47.27 | 58.29 | 64.67 | 32.30 | 30.38 | 46.38 |
| 110k | SFT | 65.40 | 82.52 | 71.28 | 68.65 | 81.91 | 36.48 | 59.31 | 71.55 | 37.21 | 34.30 | 49.54 |
| 110k | RL | 72.41 | 93.16 | 84.24 | 89.32 | 82.58 | 51.19 | 64.70 | 82.02 | 25.14 | 24.58 | 38.10 |
| 110k | SFT+RL | **77.12** | **96.62** | **89.83** | **93.25** | **86.80** | **56.33** | **71.40** | **86.63** | **43.54** | **38.97** | **55.23** |

## A.4 END-TO-END PLANNING RESULTS ON THE METAAD DATESET

We also evaluate the effectiveness of AlphaDrive for end-to-end trajectory planning on the MetaAD dataset, as summarized in Tab. 7. The comparison models include the end-to-end model VADv2, the vision-language model Qwen2VL, and an SFT-trained baseline that shares the same architecture as AlphaDrive. We employ Qwen2VL as the VLM and VADv2 as the end-to-end module to ensure a fair comparison.

The results indicate that directly using Qwen2VL for trajectory planning yields poor performance. Compared to the standalone end-to-end model, the SFT-trained baseline achieves moderate improvement by combining the VLM and end-to-end modules. Notably, AlphaDrive achieves the best planning performance among all evaluated methods, which demonstrates the effectiveness of AlphaDrive's training strategy for driving planning.

## A.5 ABLATION STUDY ON TRAINING DATASET SIZE

Fig. 1 illustrates the impact of different training data size and strategies on overall planning accuracy, while Tab. 8 provides a more detailed analysis. As observed, when the training data size decreases, SFT is more affected. With only 20k samples, the model trained with RL reaches a planning accuracy of 46.08%, which is significantly higher than that of the SFT-trained model. When using nearly half of the data, with 50k samples, AlphaDrive already achieves a planning accuracy of 70.83%, demonstrating the efficiency of our training strategy.

## A.6 MORE DATASET DETAILS

The MetaAD dataset was collected by expert human drivers using a sensor suite that includes six surround-view cameras, one fisheye camera, and a LiDAR unit, while also recording navigation information and the ego odometry at each time step. The acquisition frequency is 2 Hz. After the raw data are collected, object bounding boxes and other annotations are generated using an offline, cloud-based labeling system.

The final collection encompasses a range of weather conditions, including sunny, cloudy, and rainy days. It was captured in diverse environments such as urban areas, rural regions, and elevated highways, and provides a balanced distribution of various decision-making scenarios.

