# OpenReview forum: "AlphaDrive: Unleashing the Power of VLMs in Autonomous Driving via RL and Reasoning"
_ICLR.cc/2026/Conference — ICLR 2026 Conference Withdrawn Submission_

### Official Review · Reviewer_ched · 2025-10-28

**Soundness:** 2
**Presentation:** 2
**Contribution:** 2
**Rating:** 2
**Confidence:** 5

**Summary:**

AlphaDrive integrates GRPO based RL with VLMs to be used in autonomous driving. They employ a two stage planning reasoning training strategy combining SFT with RL. They use a distillation strategy to train a small VLM to approximate the output of a larger reasoning model as their SFT. Then, for the RLFT, they propose four reward functions and learn using GRPO. Their results show clear benefit of GRPO over SFT and ablation studies show the importance of their reward design and sample efficiency.

**Strengths:**

1. Application of GRPO in autonomous driving is quite novel, with improvements over simple SFT. This could counter some of the issues with imitation learning and possibly help with emerging reasoning ability.
2. Reward shaping allows for more flexible training paradigm with smaller data in post training fashion. This opens up the possibility of training with specifics of the operational design domain like training for trucks vs cars or urban vs highway etc.

**Weaknesses:**

1. RL is done offline. While the idea of GRPO is appealing, but the execution can be improved by using a closed loop simulator like CARLA etc. and rolling out candidate policies to calculate rewards.
2. Reproducibility. The reasoning data generation is manually curated, this makes it hard to reproduce and expensive.
3. Discrete rewards. The main reward is F1 score to measure the quality of the trajectory taken. This is a continuous action, hence a distance measure would be better fit.
4. Use of non-reactive benchmark. Navsim is non reactive, so this does not reflect closed loop performance in simulation. The benchmarks should have been bench2drive or carla based.
5. Comparison baselines. The methods compared against in Table 2 are mostly non VLM based. However, it is more appropriate to compare against VLA methods for this work like Similingo etc.

**Questions:**

1. What is MetaAD, why is this being used as a benchmark without source?
2. What are the rules for discretizing the continuous trajectory? Is there an ablation study with sensitivity of this?
3. What are the runtime and memory requirements of such a system, there should be some mention of the practicality of a pure VLM based driving model for inference.

---

### Official Review · Reviewer_fJ4n · 2025-10-31

**Soundness:** 3
**Presentation:** 3
**Contribution:** 2
**Rating:** 4
**Confidence:** 5

**Summary:**

This paper proposes AlphaDrive, a framework that trains a Vision-Language Model (VLM) for high-level autonomous driving planning using a SFT + RL strategy. The core contribution is the application of Group Relative Policy Optimization (GRPO) with four planning-specific rewards (accuracy, action-weighted, diversity, and format) to train the model to generate meta-actions and an accompanying reasoning process.

**Strengths:**

1. The method demonstrates performance gains, achieving improvements in planning accuracy on the MetaAD dataset and end-to-end planning PDMS scores on the NAVSIM benchmark.
2. The paper presents a well-motivated application of GRPO to autonomous driving. The design of the four custom rewards does make sense.
3. The proposed training strategy shows data efficiency, providing a more performance boost in low-data regimes compared to SFT alone.

**Weaknesses:**

1. The main innovation of the paper is to use RL for the high-level meta command. However, this paradigm may lack significance because the meta command has lower granularity and may not help with edge case scenarios and safety-related cases. The author is encouraged to provide analysis on how this approach may have better performance for the challenging scenario, especially in end-to-end performance.
2. I didn't see any references in the paper about the MetaAD dataset. The author is expected to provide references or provide a discussion and analysis on why the dataset is meaningful and significant if it is part of the contribution of the paper.
3. The author claims that AlphaDrive improves both performance and training efficiency. However, it lacks direct evidence on how it improves the training efficiency. Adding reasoning steps and extra training steps would usually reduce the training efficiency per data sample.

**Questions:**

1. How do you choose your meta actions, what is the semantic meaning of the "keep"?
2. How the dataset is constructed, and what the "reasoning step" looks like. Is the reward signal only relevant to the final planning action, or does it have intermediate supervision (like critical object, segmentation, etc.)?

---

### Official Review · Reviewer_h7AT · 2025-11-01

**Soundness:** 3
**Presentation:** 4
**Contribution:** 3
**Rating:** 6
**Confidence:** 4

**Summary:**

This paper presents a reinforcement-learning based reasoning framework for end-to-end action planning in autonomous driving. The goal is to benefit from reasoning techniques of VLMs and include RL for improved driving policies. Specifically, the authors integrate GRPO and a VLM with a focus being on the developed rewards. The paper presents a very performant distilled driving network and show that their VLM can provide multiple valid solutions for a driving situation.

**Strengths:**

The paper presents an interesting knowledge distillation method from a cloud based VLM, which would not be real-time capable in the near future, to a fine-tuned model that is trained with traditional supervision.

The paper is for the most part very clear. The pseudo-algorithm helps understanding the different rewards.

The performance on the MetaAD validation set, even without fine-tuning on it, is very impressive.

The approach seems to have a clear edge upon other approaches on the NAVSIM dataset.

Generating multiple valid options for the same situation is pretty remarkable. While this can be observed for general trajectory planning approaches, the reasons the model gives shows the correct motivations for the different action alternatives.

The paper ablates the method in detail in Table 3 and find useful engineering tricks like a warm-up phase, described in section 3.3

**Weaknesses:**

If the authors of this work are affiliated with Meta I would strongly suggest to re-name the dataset during the review process. If not then the name is at least peculiar. The dataset is either not cited properly or self-collected. This is not clear from this paper which describes general features of the dataset but not if the authors collected it or not. The language used says they "adopt" MetaAD but I am not aware of this dataset and a search did not deliver any results. This should be clearer and ideally, the dataset should be made available to the research community.

It is hard to take insights from the video in the supplementary material. It seems the vehicle is driving more or less according to the predictions but it is very sped up and shows diverse scenes. It would have been better taking maybe one-two scenes, showing them slower and overlay some text to explain something concrete the model mastered or not.

In summary, the approach seems very clearly presented and well evaluated. Including a VLM into an end-to-end driving approach like this seems very promising to benefit from the information from large models while having a perspective to be real-time capable. One big downside is the confusion around the MetaAD dataset, where it comes from, what is the split size, why wasn't the test split used for Table 1 etc. While results were compared on the NAVSIM dataset, the performance is about to saturate and that dataset may no longer be suitable to see meaningful differences.

**Questions:**

In figure 2 the model answer is a left_turn. What are the actions the language model can give? If the model wants to avoid pedestrians but go straight as prompted, it should probably do a slight left turn, swerve left or output something similar but not do an actual turn. So, what are valid outputs of the model?

Why were the results in Table 1 compared on the validation and not the test set?

Who made the MetaAD dataset? How big is the train/val/test split? Will it be published?

The caption in Figure 3 says "AlphaDirve".

---

### Official Review · Reviewer_1ogH · 2025-11-01

**Soundness:** 3
**Presentation:** 2
**Contribution:** 3
**Rating:** 6
**Confidence:** 5

**Summary:**

This paper presents AlphaDrive, which is a reinforcement learning (RL), and a reasoning framework for vision-language models (VLMs) to be used in autonomous vehicles. This paper identifies the limitations of using only supervised fine-tune methods (SFT) to train VLMs; therefore the authors present a two stage training method that uses SFT and Group Relative Policy Optimization (GRPO) based RL. In addition, the authors define four new "planning oriented" rewards—accuracy, action weighted, diversity and format —to assist in aligning the rewards with the needs of driving planning tasks. Results from experiments performed using data from the MetaAD and NAVSIM dataset provide evidence of large increases in both the accuracy and efficiency of driving planning and also evidence of emergent multimodal planning capabilities. As such the authors present AlphaDrive as an advancement toward integrating reasoning with RL in the development of autonomous vehicles.

**Strengths:**

-GRPO-based reinforcement learning combined with VLMs to aid in planning is new and well motivated.
- Four rewards used in this study were a good design for both safety critical and multi-modal tasks.
- A two stage training regime (SFT warm up followed by RL) was shown to improve both the convergence rate of the learning process and the stability of the reasoning process.
- Empirical results that showed improvement in planning accuracy and reasoning quality on two major public datasets.
- Emergent multimodal planning behavior demonstrated in this study and shows promise for developing adaptive planning systems in the future.

**Weaknesses:**

- While this system is capable of utilizing pseudo labeled reasoning data from large language models (i.e. Qwen2VL-72B) it can also lead to bias and/or noise in the data.
- Evaluation in this study was very robust but there was no direct testing of the system in a real world environment, only simulated environments.
- There is no discussion in the study about potential increased computational costs associated with using this method and what would be required for its feasible use in real time driving systems.
- Figures and tables are sometimes presented at high density levels and could have been presented in a more clear fashion.

**Questions:**

1) Can you describe how much of an effect would it have on your results if you were to use a different "large" model to generate the reasoning data for AlphaDrive?
2) Can you provide a quantification of the increased computational cost associated with training with GRPO relative to both PPO and DPO?
3) Did the authors evaluate AlphaDrive's ability to generalize to new (unseen) roads and/or sensor noise?
4) Do you believe that the AlphaDrive model can be executed in real time, given the current hardware constraints of an onboard driving computer?

---

### Note · Authors · 2025-11-13

I have read and agree with the venue's withdrawal policy on behalf of myself and my co-authors.